# Magnetically tunable bidirectional locomotion of a self-assembled nanorod-sphere propeller

José García-Torres[1,2,3], Carles Calero[1,3], Francesc Sagués[2,3], Ignacio Pagonabarraga [1,4,5] & Pietro Tierno [1,3,4]

Field-driven direct assembly of nanoscale matter has impact in disparate fields of science. In microscale systems, such concept has been recently exploited to optimize propulsion in viscous fluids. Despite the great potential offered by miniaturization, using self-assembly to achieve transport at the nanoscale remains an elusive task. Here we show that a hybrid propeller, composed by a ferromagnetic nanorod and a paramagnetic microsphere, can be steered in a fluid in a variety of modes, from pusher to puller, when the pair is dynamically actuated by a simple oscillating magnetic field. We exploit this unique design to build more complex structures capable of carrying several colloidal cargos as microscopic trains that quickly disassemble at will under magnetic command. In addition, our prototype can be extended to smaller nanorods below the diffraction limit, but still dynamically reconfigurable by the applied magnetic field.

[1] Departament de Física de la Matèria Condensada, Universitat de Barcelona, 08028 Barcelona, Spain. [2] Departament de Ciència de Materials i Química Física, Universitat de Barcelona, 08028 Barcelona, Spain. [3] Institut de Nanociència i Nanotecnologia, Universitat de Barcelona, 08028 Barcelona, Spain. [4] Universitat de Barcelona Institute of Complex Systems (UBICS), Universitat de Barcelona, 08028, Barcelona, Spain. [5] CECAM, Centre Européen de Calcul Atomique et Moléculaire, École Polytechnique Fédérale de Lasuanne, Batochime, Avenue Forel 2, 1015 Lausanne, Switzerland. These authors contributed equally: José García-Torres, Carles Calero. Correspondence and requests for materials should be addressed to P.T. (email: ptierno@ub.edu)

Self-assembly at the nanoscale induced by an external field has a direct impact on different fields[1,2], from the rapid fabrication of photonic lattices[3–5], to the manipulation of liquid droplets[6,7], DNA structures[8,9], colloidal[10], or molecular[11] networks. Propelling micro and nanomachines in fluid media have the potential to revolutionize different emerging technologies, since they can be used as efficient drug-delivery vectors[12,13] in microfluidic[14] and biological[15] networks, as non-invasive microsurgey vehicles[16] or as chemical biodetectors[17]. In order to realize these challenging tasks, a robust and adaptable swimming strategy is required based on the knowledge of the physics of fluid flows at low Reynolds number, where these prototypes operate. Under such conditions, viscous forces dominate over inertial ones, and the corresponding hydrodynamic laws are time-reversible[18]. A direct, non-trivial consequence of this time reversibility is that any immersed object displaying a reciprocal motion, namely a symmetric backward and forward shape distortion, will not generate net motion[19]. This situation can be avoided in different ways, for instance by designing micropropellers that can be actuated by external fields[20–25], or that can act as catalizers of a chemical reaction on their surface[26,27]. Following such strategies, most of the realized prototypes are single-component objects that do not require the interaction with a neighbor element to move within the medium. Modular prototypes where different parts interact and cooperate, may significantly advance the state-of-the-art in the field enabling compartmentalization, high payload due to more surfaces available and stability against external perturbations[28]. While recent experiments with ferromagnetic particles demonstrated the potential for such approach[29–34], nanoscale multicomponent prototypes remain challenging due to the required complex balance between the different forces that arise at such scale.

In this article, we report the realization of a hybrid nanorod-sphere propeller that can be manipulated and transported in viscous fluids via application of a simple in-plane time-dependent magnetic field. The external drive induces a synchronized cooperative movement between the two elements that periodically attract and repel due to dipolar forces. The periodic relative displacement induces a fast rotational motion of the nanorod during each field cycle, that is rectified into a net translation due to the close proximity of a bounding plate. Our velocity tunable magnetic prototype allows to switch the propulsion direction by varying either the amplitude or the frequency of the applied field. Thus, we show both via experiments and numerical simulations that two different, interchangeable mechanisms of motion may be selected upon balancing magnetism, gravity, and hydrodynamics. Beyond the achievements of single-component microswimmers, our system allows: (i) independent and selective control of one of the two elements of the dimer, (ii) collective transport of several colloidal cargos via attractive dipolar forces, and (iii) further downscaling the nanorod size, a feature that broadens the spectrum of nanoscale objects that can be transported in a liquid medium.

## Results

**Bidirectional motion of the hybrid propeller.** Our magnetically assembled propeller is composed by a paramagnetic spherical particle of radius $a = 1.5\ \mu m$ and a ferromagnetic nickel nanorod of length $l = 3\ \mu m$ and diameter $d = 400\ nm$. Both elements display different magnetic properties under an external field $\mathbf{B}_{ext}$. The particle acquires a tunable dipole moment $\mathbf{m}_p = \frac{4\pi}{3\mu_0}a^3\chi\mathbf{B}_{ext}$, which aligns with the field. Here $\mu_0 = 4\pi \times 10^{-7}\ Hm^{-1}$ and $\chi$ denotes the magnetic volume susceptibility (see Methods for the determination of $\chi$). In contrast, the nickel nanorod has a fixed, permanent moment $\mathbf{m}_n$ oriented along its long axis, as measured in separate experiments (Methods).

To propel the hybrid prototype, we apply an in-plane square wave modulation, characterized by an amplitude $B_0$ and a frequency $\nu$, $\mathbf{B}_{ext} = B_0\ \text{sgn}(\sin(2\pi\nu t))\mathbf{e_x}$, where sgn denotes the sign function. This simple actuation scheme will not induce any translational motion to the separate elements of the pair, even when placed close to a boundary wall. Specifically, the magnetic field does not lead to any motion on the colloidal sphere and it exerts a torque on the nanorod, inducing a reciprocal motion characterized by a back and forth nanorod rotation around its

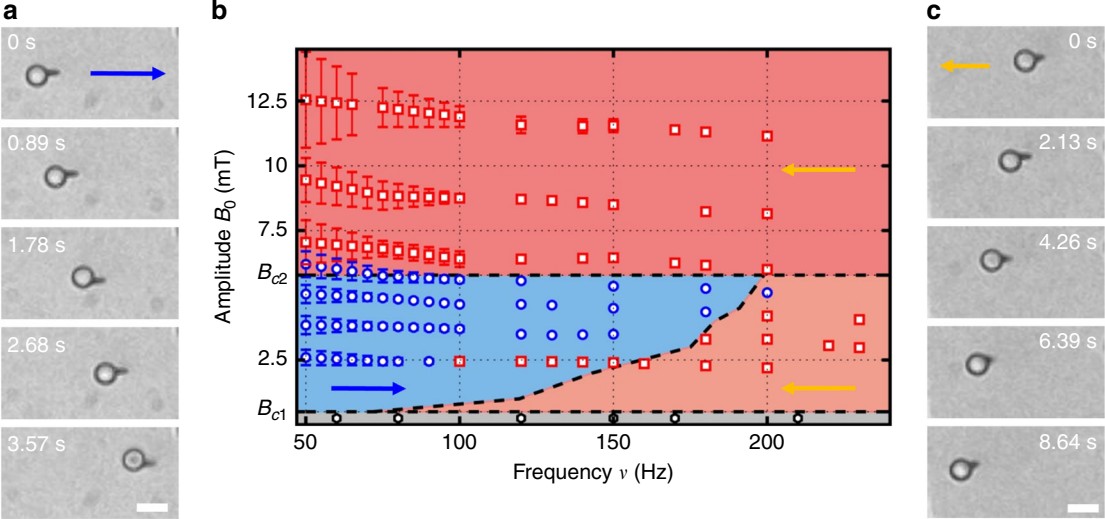

**Fig. 1** Bidirectional transport of the sphere-nanorod prototype. **a, c** Sequences of images of the assembled pair propelling right with the nanorod at the front (**a**), and left with the nanorod at the back (**c**). In **a, c**, the in-plane square wave field has frequency $\nu = 100$ Hz and amplitude $B_0 = 4.2$ mT (**a**) (and $\nu = 100$ Hz, $B_0 = 8.9$ mT (**c**)). The scale bar is 5 μm for both sequences. See Supplementary Movie 1. **b** Diagram in the ($\nu$, $B_0$) plane illustrating the regimes where propulsion with the nanorod in front (blue region) and behind (two red regions) the spherical colloid is observed. In the diagram, the threshold fields are $B_{c1} = 0.5$ mT and $B_{c2} = 5.8$ mT. The dashed lines delimiting the different regions in the diagram are obtained from numerical simulation, while scattered points are experimental data. Error bars in the data reflect the decay of the applied signal within each semicycle of the square wave field. They are given by the difference between the maximum and minimum values of the applied magnetic field in one semicycle

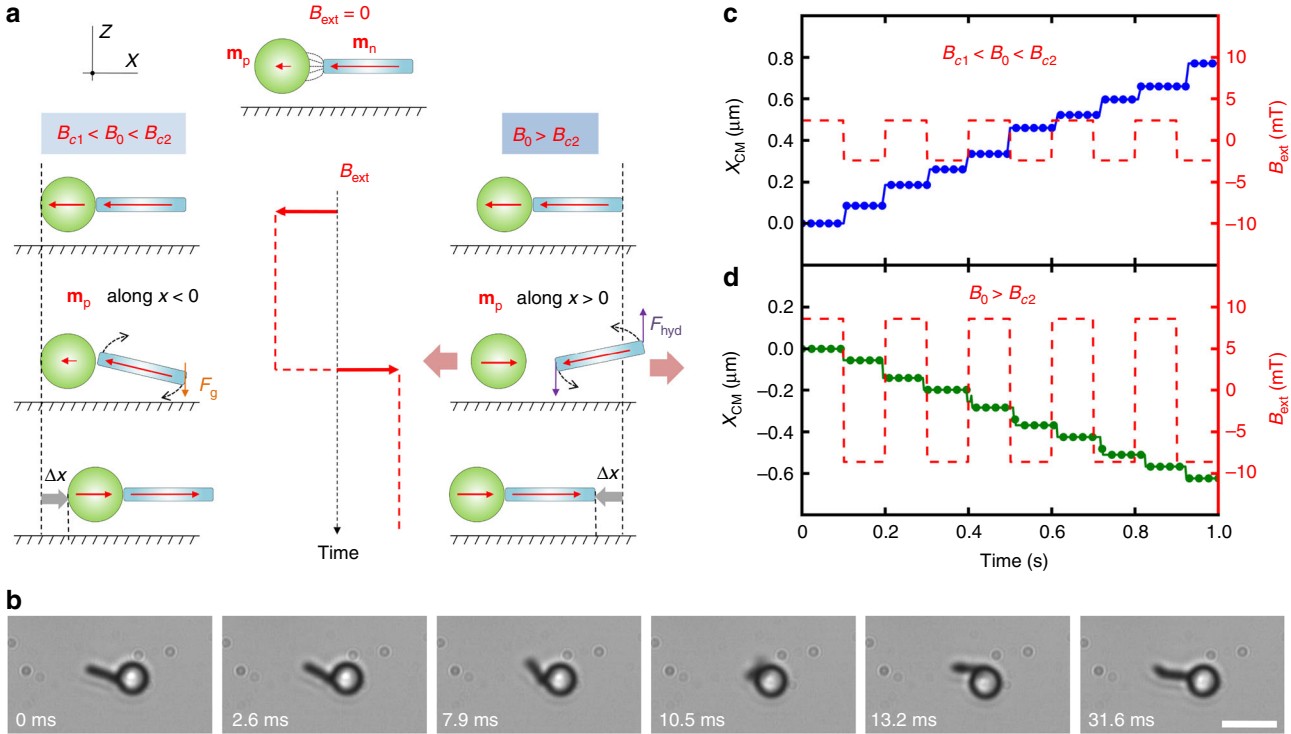

**Fig. 2** Transport mechanisms. **a** Sequence of schematics in the $(x, z)$ plane illustrating the two transport modes that depend on the amplitude $B_0$. In the first column ($B_0 < B_{c2}$) the gravitational force $F_g$ induces a negative tilt angle of the nanorod, forcing the latter into a clockwise rotation when the applied field changes direction, and thus the nanorod drags the spherical colloid. In the second column ($B_0 > B_{c2}$), the induced moment $\mathbf{m}_p$ in the paramagnetic colloid follows the applied field, and dipolar repulsion between the pair separates the rod away from the paramagnetic colloid. Hydrodynamic interaction with the surface $F_{hyd}$ induces a positive tilt angle that triggers counterclockwise rotation of the nanorod, and the nanorod pushes the spherical colloid. **b** High-speed camera images illustrating the rotational motion of the nanorod during one field reversal ($B_0 = 2.6$ mT, $\nu = 4$ Hz). Scale bar is 5 μm, see Supplementary Movie 2. **c, d** Graphs illustrating the center of mass position of the prototype plotted versus time corresponding to the situations $B_{c1} < B_0 < B_{c2}$ (**c**) and $B_0 > B_{c2}$ (**d**), and measured under an applied field with frequency $\nu = 5$ Hz. Dashed red line denotes the amplitude of the applied magnetic modulation

center. However, when close to each other, the constituents of the pair interact via dipolar forces that give rise to a periodic series of attraction-repulsion sequences. These cycles trigger a chiral rotation of the nanorod as it follows the field dynamics. Within each cycle, the nanorod rotational motion is rectified into a net translation due to the close proximity of a bounding plate and the generated asymmetry in the frictional force experienced by the nanorod[21,22]. Thus, the directed motion of the pair results from a symmetry breaking mechanism induced by the coordinated interaction of its constituents. In contrast to other magnetic prototypes composed by a single unit, here the propulsion direction can be switched by varying either $B_0$ or $\nu$. As shown in Fig. 1a, c, the nanorod can either act as a puller, by dragging the paramagnetic sphere (a), or as a pusher by translocating the particle in front of it (c). The two directions result from different propulsion mechanisms, involving distinctive balances between the relevant competing forces, either: magnetism and gravity (a), or magnetism and hydrodynamics (c). The state diagram in the ($\nu$, $B_0$) plane, Fig. 1b, shows that the first mechanism applies for intermediate magnitudes of $B_0$, between the thresholds $B_{c1}$ and $B_{c2}$, and at small frequencies, while below $B_{c1}$ the pair did not show any net motion, and thermal fluctuations dominate the displacement of the pair.

**Propulsion mechanisms**. To elucidate the propulsion mechanism of the hybrid prototype, we measured high-speed videos during its displacement (Supplementary Movie 2), and confirm that for both mechanisms, the nanorod rotates around its center of velocity while following the switching of the applied field (Fig. 2a,

b). The characteristic reorientational time of the nanorod under a magnetic field of magnitude $B_0$ is given by $\tau_{rot}(B) = \xi_r / (B_{ext} m_n)$, where $\xi_r$ is its rotational friction coefficient. For the relevant magnitude ranges used in the experiments, $\tau_{rot} \sim 10^{-5}$ s–$10^{-4}$ s, thus much shorter than the employed oscillation period ($\sim 10^{-2}$ s). Consequently, as shown in Fig. 2b, the ferromagnetic rod rotates every semi-period of oscillation to align itself with the applied field.

The series of schematics in Fig. 2a illustrate the two mechanisms of motion that arise during one reversal of the applied field. In absence of field, as shown in the top image, the paramagnetic colloid becomes magnetized by the field created by the nanorod, and there is a weak attraction between the two elements. Under an external field $\mathbf{B}_{ext}$, the correspondingly induced moment in the spherical colloid is given by $\mathbf{m}_p = \frac{4\pi}{3\mu_0} a^3 \chi$ ($\mathbf{B}_{ext} + \mathbf{B}_{ferro}$), where $\mathbf{B}_{ferro}$ is the magnetic field created by the nanorod on the center of the paramagnetic colloid. When the nanorod is aligned with the external magnetic field (and, thus, with $\mathbf{m}_p$), both particles attract due to dipolar forces. During field reversal and before the nanorod reorients, their mutual interaction depends on the amplitude of $\mathbf{B}_{ext}$. For $|\mathbf{B}_{ext}| < |\mathbf{B}_{ferro}|$, $\mathbf{m}_p$ remains aligned with $\mathbf{m}_n$ and both particles still attract. As shown in the first column of Fig. 2a, the tilt of the nanorod arises from the combined action of magnetic attraction, that decays along the rod with the distance to the spherical particle, and gravity, which acts homogeneously along the rod. Such misalignment with the horizontal prompts a clockwise (counterclockwise) rotation of the nanorod to align its magnetic moment $\mathbf{m}_n$ with $\mathbf{B}_{ext}$ if the spherical particle is located on the left (right). As a consequence

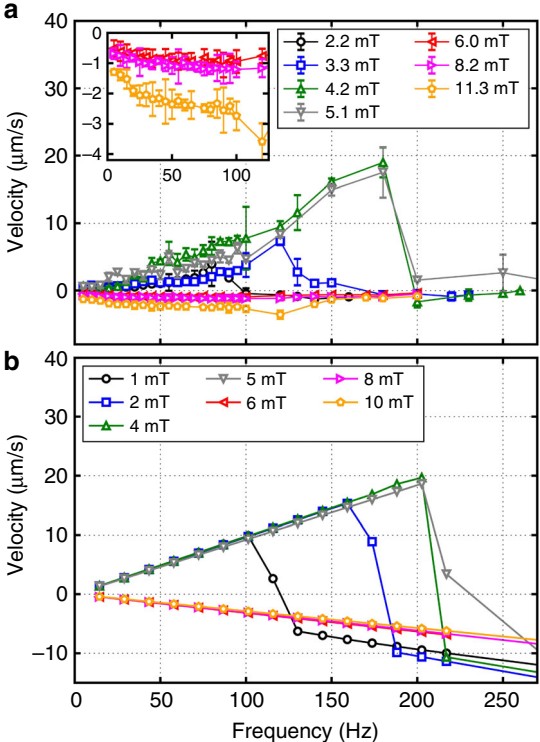

**Fig. 3** Average speed of the hybrid propeller. **a**, **b** Comparison between the experimental data (**a**) and the numerical results (**b**) for the average velocity of the composite propeller versus frequency of the applied field $\nu$. The inset in the top graph shows an enlargement of the experimental data when the nanorod acts as a pusher. In both cases, the curves are taken at different amplitudes $B_0$ of the applied field. Positive velocities describe the transport when the nanorod pulls the spherical colloid, while negative velocities denote the regime where the nanorod acts as a pusher. Error bars in **a** are the standard deviation of the measurements

of its rotational motion, the nanorod translates due to the close proximity of the surface, see the geometry in Fig. 2a. The hydrodynamic interaction with the surface creates an asymmetry in friction that is rectified into a net translational motion[35]. Thus, the nanorod acts as a puller by dragging the spherical colloid along, see Supplementary Movie 1.

When $|\mathbf{B}_{ext}| > |\mathbf{B}_{ferro}|$, $\mathbf{m}_p$ becomes antiparallel to $\mathbf{m}_n$ when the field switches, and the nanorod and the particle repel each other, as shown in the second column in Fig. 2a. As the two particles move away from each other, they generate a hydrodynamic flow whose vertical direction is non-uniform along the ferromagnetic rod. Thus, the nanorod experiences a hydrodynamic torque with the free tip directed upward, see Supplementary Fig. 1 and Supplementary Note 1 for a formal derivation. As a consequence, the nanorod rotates in a counterclockwise (clockwise) direction if the spherical colloid is on its left (right) to align its magnetic moment with the external field. Due to such rotation, the nanorod acts as a pusher, translocating the paramagnetic colloid in front of it, see Supplementary Movie 1.

These two mechanisms explain the different regimes of motion identified in Fig. 1b. Moreover, the threshold value of the magnetic field obtained experimentally is $B_{c2} = 5.8$ mT, in good agreement with the calculated value of the magnetic field created by the ferromagnetic nanorod at the center of the paramagnetic colloid $B_{ferro} \approx 5$ mT. In Fig. 2c, d, the trajectory of the hybrid propeller is shown for field amplitudes below and above $B_{c2}$ at a fixed frequency, evidencing the inversion of the direction of motion. In both cases, the discrete jumps reflect the reversal of

the external field. The two regimes identified rely on the effective dipolar interaction between the ferromagnetic nanorod and the paramagnetic colloid as the nanorod rotates to realign with the oscillating magnetic field. Therefore, not only the magnitude and frequency of the applied magnetic field, but also the shape of the signal affects the nature and mode of the pair self-propulsion. For example, we have experimentally verified that for a sinusoidal magnetic field, the magnetic swimmer acts as a puller independently of the field amplitude since the external field reverses direction continuously and the nanorod thus reorients under fields smaller than $B_{c2}$. Hence, one can also use the time dependence of the forcing signal to control the nature and performance of the motion of the self-assembled magnetic swimmer.

At high frequencies, our simulations reveal an additional swimming regime in which the nanorod pushes the spherical colloid independently of the field amplitude, i.e., the red region enclosed by fields $B < B_{c2}$ in Fig. 1b. The presence of a bounding wall induces a hydrodynamic lift of the nanorod, which subsequently sediments back to the dimer equilibrium configuration in a characteristic time $\tau_r$. The relaxation time can be expressed as a sum of two contributions, $\tau_r = \tau_{sed} + \tau_{align}$. Here, $\tau_{sed}$ is the sedimentation time that takes the propeller to make contact with the bounding plane after the rotation of the nanorod, $\tau_{sed} \approx 6\pi\eta a h/(M_p g)$, where $h$ is the distance the sphere is lifted as a result of the rotation of the nanorod, $\eta$ is the viscosity of the fluid, $g$ the acceleration of gravity, and $M_p$ the mass of the spherical colloid. $\tau_{align}$ is the time to align $\mathbf{m}_n$ with $\mathbf{B}_{ext}$ from the orientation adopted by the nanorod during sedimentation, $\tau_{align} \approx 2\xi_r^p/(m_n B_{ext})\log(C/(m_n B_{ext}))$, being $\xi_r^p$ the rotational friction coefficient of the propeller and $C$ a fitting constant (see Supplementary Note 2, Supplementary Figs. 2 and 3). If the field's oscillating frequency $\omega$ is higher than a critical value $\omega_{max} = 2\pi/\tau_r$, the pair cannot relax to the equilibrium configuration, breaking the cyclic motion described above. Instead, for $\omega > \omega_{max}$, the rotation of the ferromagnet is dominated by the orientation that the nanorod adopts in the sedimentation of the pair, with the free tip of the nanorod directed upward due to the lower friction per mass of the spherical colloid. Consequently, when the external field reverses direction, the nanorod rotates counterclockwise (clockwise) if the paramagnetic colloid is on its left (right) to align its magnetic moment with the external field, see also Supplementary Movies 6 and 7.

**Propeller speed**. In Fig. 3, we plot the average speed of the magnetic propeller for different values of the amplitude and frequency of the applied field, as obtained in experiments (Fig. 3a) and simulations (Fig. 3b). At low frequencies, the average speed increases linearly with the oscillating frequency, consistent with the mechanism of motion by steps generated by the rotation of the nanorod. The asymmetry in the magnitude of the velocity when acting as a puller or pusher is attributed to the different contribution to the propeller motion from the rotation of the paramagnetic colloid. For magnetic fields with $B_0 > B_{ferro}$ (pusher), the change of direction of the magnetic moment of the paramagnet $\mathbf{m}_p$ occurs along its easy axis when the field changes polarity and no rotation of the spherical colloid takes place. In contrast for $B_0 < B_{ferro}$ (puller), the direction of $\mathbf{m}_p$ follows the reorientation of the nanorod, which induces a rotation of the spherical colloid enhancing propulsion. The rotational motion of the paramagnetic sphere results from the presence of magnetic anisotropy within the particle. Indeed, in a separate experiment, we have observed that under a rotating field circularly polarized in the $(x, z)$ plane, the particle experiences a magnetic torque, and acquires a net translational motion close to the glass surface.

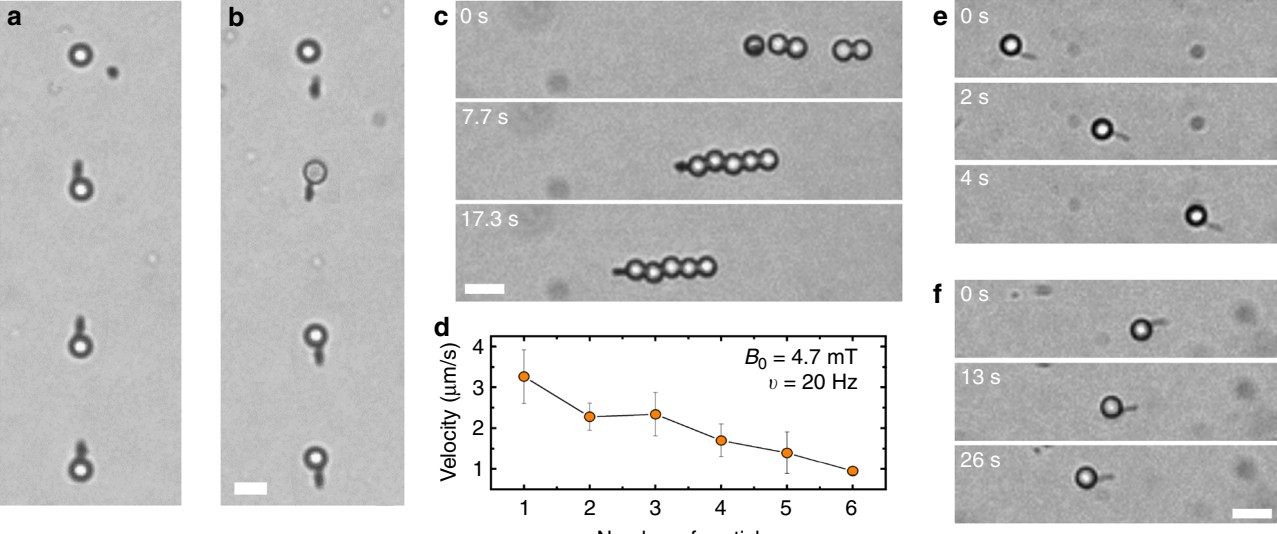

**Fig. 4** Reconfiguration, cargo transport, and miniaturization. **a**, **b** Superimposed sequence of images showing the controlled release (**a**) and pickup (**b**) of a spherical colloid by the propelling nanorod ($B_0 = 3.4$ mT, $\nu = 20$ Hz). In the first case, the pair was separated by applying a perpendicular field $B_z = 8$ mT that induces repulsive dipolar interactions, see Supplementary Movie 3. **c** Sequence of images demonstrating the multicargo transport by one nanorod via attractive dipolar forces ($B_0 = 4.7$ mT, $\nu = 20$ Hz) (Supplementary Movie 4). **d** Average speed of the train versus number of particles in the chain. Error bars are the standard deviation of the measurements. **e**, **f** Transport of the microparticles according to the two mechanisms by a thinner nanorod having length $l = 3.0$ μm and diameter $d = 100$ nm. **e** shows locomotion with nanorod in front ($B_0 = 4.7$ mT, $\nu = 20$ Hz), **f** with nanorod on the back ($B_0 = 5.8$ mT, $\nu = 20$ Hz). See corresponding Supplementary Movie 5. Scale bars in all images are 5 μm

At high frequencies, $\omega > \omega_{max}$, the propeller acts as a pusher independently of the field amplitude. For cases with $B_0 < B_{ferro}$, this entails a sudden reduction of the velocity and a reversal of its direction for $\omega > \omega_{max}$. We note that, when comparing the two graphs in Fig. 3, the experimental data display some deviations from the simulations especially when the nanorod acts as a pusher, here characterized by a smaller translational velocity. These deviations can be attributed to the presence of noise in the experimental system, coming, e.g., from thermal fluctuations or inhomogeneities in the glass surface and in the magnetic moment of the nanorod. Despite that, in this regime, we found a robust trend where the velocity of the ensemble reverses its direction as the frequency increases, as shown in the inset on top of Fig. 3.

## Discussion

The developed hybrid nanorod spherical colloid propeller shows exceptional functionalities that may open new possibilities to explore a wide range of applications. For example, the transport of magnetic cargoes with controlled directionality and the possibility to disassemble the self-propelled pair at any convenient time may find direct application in lab-on-a-chip devices. In particular, we show in Fig. 4a (Supplementary Movie 3) the possibility to translocate and leave the spherical colloid at a desired location of the experimental platform. The disassembly was obtained by simple application of a perpendicular field that aligns the nanorod, while inducing a parallel dipole moment to the paramagnetic colloid. This configuration is controlled by repulsive dipolar interactions that break the pair apart releasing the particle in the fluid. This operation may be repeated at will, by re-assembling the pair and transporting the colloidal cargo to a different place, as shown in Fig. 4b. Another future possibility would be to chemically functionalize the spherical colloid in order to transport a chemical or biological cargo to a target place where a biochemical reaction may take place.

We further show in Fig. 4c that such a simple concept may be extended to several particles that can be collectively transported and released at a given place (see also Supplementary Movie 4).

Once the train of particles is formed, the single nanorod is able to drag it at a constant speed along a direction of motion that depends on the corresponding field parameters which select the given transport mechanism. As shown in Fig. 4d, at fixed amplitude and frequency of the applied field, the average speed of the train decreases almost linearly with the number of particles in the chain, due to the corresponding approximately linear increase in the friction coefficient. Furthermore, the motion of the entire self-assembled structure may be reversed by increasing the amplitude of the applied field.

We finally demonstrate the possibility to further miniaturize the system below the optical resolution limit, by using a thinner nanorod having 100 nm in diameter (Fig. 4e). Such nanorods were prepared using the same methodology with a membrane characterized by smaller pores, and assembled using similar protocols than in previous experiments. As shown in Fig. 4e, f, we can recover the two different mechanisms of motions, with the nanorod acting as a puller (Fig. 4e) or acting as a pusher (Fig. 4f). As it can be observed from the sequences of images, the forward motion also in this case is faster than the reverse one. This confirms the generality of our approach to propel micro and nanoscale objects, thus covering a wide range of length scales, and based on the cooperative magnetic assembly of the pair.

## Methods

**Experimental details**. Nickel nanowires were synthesized via template-assisted electrode position in a 0.1 M $NiCl_2$ plating solution (Sigma Aldrich) and using polycarbonate (PC) membranes (Millipore) as a template. One side of the PC membrane was coated with a thin gold layer and used as working electrode. The counter electrode was a platinum foil and the reference electrode was Ag/AgCl/KCl (3M). The length and the diameter of the growing nanowires were varied by adjusting the electrodeposition time and the pore size in the PC, respectively. The nanorods were released by, first removing the gold layer with a $I_2/I^-$ saturated solution and then by wet etching of the PC membrane in $CHCl_3$. The nanowires were then washed with chloroform (ten times), chloroform-ethanol mixtures (three times), ethanol (two times), and deionised water (five times). Finally, the nanorods were dispersed in a solution containing sodium dodecyl sulphate. Structural and morphological investigations were carried our with scanning electron micro-scopy and high-resolution transmission electron microscopy.

As spherical colloids, we use paramagnetic microspheres with diameter 3 μm, ~15% iron oxide content and surface carboxylic groups (ProMag PMC3N, Bang Laboratorie). The particles and the nanorods were dispersed in highly deionized water (MilliQ, Millipore) and let sediment above a glass plate. The plate was placed in the center of two orthogonal coils arranged on the stage of a light microscope (Eclipse Ni, Nikon). The coils are connected to a wave generator (TGA1244, TTi) feeding a power amplifier (IMG AMP-1800), and particle dynamics were recorded with charge coupled device cameras working at 75 fps (scA640-74fc, Basler) or 380 fps (Ximea). In our system, we work under dilute conditions (~200 nanorods per ml and ~200 spheres per ml) in order to focus on the dynamics of isolated propellers. However for the train formation, larger quantities of particles were used, ~500 spheres per ml.

**Determination of the magnetic moments.** Nanorods: We measured the permanent moment of the Nickel nanorods $m_n$ by following their orientation under a static external field $\mathbf{B}$. During reorientation, the magnetic torque acting on the nanorod, $\boldsymbol{\tau}_m = \mathbf{m}_n \times \mathbf{B}_{ext}$, is balanced by the viscous torque arising from its rotation in the fluid, $\boldsymbol{\tau}_v = -\xi_r \dot{\theta}$, being $\xi_r$ its rotational friction coefficient. By solving the torque balance equation, $\boldsymbol{\tau}_m + \boldsymbol{\tau}_v = 0$, we obtain, $\theta(t) = 2\tan^{-1}[\tanh(t/\tau_r)]$, where $\tau_r = 2\xi_r/(m_n B)$ is the relaxation time[36]. By using an external field $B = 0.3$ mT, we find $\tau_r = 0.04$ s and $m_n = 3.7 \times 10^{-11}$ Am$^{-2}$.

Paramagnetic colloid: We measure the magnetic susceptibility $\chi = 3\mu_0 m_p/(4\pi a^3 B_0)$ placing a pair of particles of radius $a$ in close contact and inducing repulsive interaction via a perpendicular field $\mathbf{B}_{ext} = B_0 \mathbf{e}_z$. The induced moments give rise to a repulsive magnetic force $\mathbf{F}_m = 3\mu_0 m_p^2/(4\pi \mathbf{r}^4)$ that is balanced by the viscous force $\mathbf{F}_v = 6\pi \eta a \mathbf{r}$ arising from the translational motion in water (viscosity $\eta = 10^{-3}$ Pa × s). From the force balance equation, we obtain, $r(t) = a(32 + 20B_0^2 \chi^2/(9\mu_0 \eta))^{0.2}$ and thus fitting the experimental data for $B_0 = 7.2$ mT obtain the adimensional value of $\chi = 0.21$.

**Numerical simulation.** The paramagnetic colloid is described as a sphere of radius $a$ with an induced magnetic moment $\mathbf{m}_p$ in its center. The magnetic moment of the paramagnet is determined by the action of both the external magnetic field $\mathbf{B}_{ext}$ and the field induced by the ferromagnetic rod on the paramagnet $\mathbf{B}_{ferro}$, $\mathbf{m}_p = \frac{4\pi}{3\mu_0} a^3 \chi (\mathbf{B}_{ext} + \mathbf{B}_{ferro})$. We assume an isotropic paramagnet with scalar magnetic susceptibility $\chi$. The ferromagnetic rod is modeled as a collection of $N$ equally spaced spherical beads along a straight line. Every bead carries a fixed magnetic moment $\mathbf{m}_n/N$ directed along the axis of the rod, which we assume to be its easy axis of magnetization. The paramagnetic sphere and the ferromagnetic rod interact through magnetic dipolar interactions, which in our model are calculated as a sum of dipolar interactions between the paramagnetic sphere and the $N$ beads composing the ferromagnetic rod. The magnetic particles are also subjected to a torque exerted by the external magnetic field $\boldsymbol{\tau}_B = \mathbf{m}_n \times \mathbf{B}_{ext}$.

The dimer evolves according to Newton's equations of motion, which depends on the forces acting on all the elements of the model. The non-hydrodynamic force acting on particle $i$, $\mathbf{F}_i$, is the sum of the magnetic dipolar interaction with the rest of particles, gravity (where we assume that all particles have the same density), and a repulsive, short range, steric interaction with the bounding wall, which ensures particles do not overlap with the solid bounding surface. When particle $i$ is subject to a net force $\mathbf{F}_i$, it will induce a flow field. Since the dimer moves at low Reynolds number, the flow field at a given position, $\mathbf{u}(\mathbf{r})$, is given by the superposition of the contributions from each of the particles. Assuming the components of the model dimer behave as point particles, as far as their hydrodynamic effect is concerned, the flow field at point $\mathbf{r}_i$ is given by

$$\mathbf{u}(\mathbf{r}_i) = \frac{1}{8\pi\eta} \sum_j G^{Blake}(\mathbf{r}_i; \mathbf{r}_j) \times \mathbf{F}_j, \qquad (1)$$

where $\eta$ is the viscosity of the fluid, and $G^{Blake}(\mathbf{r}_i; \mathbf{r}_j)$ is the Blake tensor[37], which ensures no-slip boundary condition for the fluid flow at the flat bounding wall. Each particle $i$ is also subject to a hydrodynamic friction force when its velocity, $\mathbf{v}_i$, differs from the local, induced fluid flow at the particle position, $\mathbf{u}(\mathbf{r}_i)$. Assuming particles can be considered as point particles in their hydrodynamic interactions, we can express the hydrodynamic friction force on particle $i$ as

$$\mathbf{F}_{H,i} = -\gamma_i(\mathbf{v}_i - \mathbf{u}(\mathbf{r}_i)), \qquad (2)$$

where $\gamma_i$ is the bead's friction coefficient.

The time evolution of the system is solved using a Verlet algorithm adapted for cases with forces that depend on the velocity[38].

**Data availability.** The data that support the findings of this study are available from the corresponding author upon request (ptierno@ub.edu). The numerical code developed in this work is also available upon request.

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

## Acknowledgements

This project has received funding from the European Unions Horizon 2020 research and innovation programme under grant agreement no. 665440. P.T. acknowledges support from the European Research Council (grant agreement no. 335040), MINECO under project FIS2016-78507-C2-2-P and Generalitat de Catalunya under project 2014-SGR-878. I.P. acknowledges support from MINECO under project FIS2015-67837-P and Generalitat de Catalunya under project 2014SGR-922. F.S. acknowledges support from MINECO under project FIS2016-C2-1-P AEI/FEDER-EU.

## Author contributions

P.T. conceived the project. J.G.-T. performed the experiments. C.C. and I.P. developed the theoretical model. P.T., F.S., and I.P. supervised the work. P.T., J.G.-T., and C.C. wrote the paper. All authors discussed and interpreted the results.

## Additional information

**Competing interests:** The authors declare no competing interests.

