## [Peer Review File · Nature Communications]

Reviewers' comments:

Reviewer #1 (Remarks to the Author):

The authors in their manuscript "Magnetically tunable bidirectional locomotion of a self-assembled nanorod-sphere propeller" demonstrate and explore in detail a new design for self-assembled and self-propelling colloidal structure that is most probably scalable to the nanoscale. Controller transport at micro and nanoscale is still a formidable challenge and each new strategy brings us a step closer to the robust and adaptive swimming at the micro and nano scales required for emergent technologies such as targeted drug delivery and soft microrobotics.

In their paper authors report realization of a hybrid nanorod-microspher propeller actuated and controlled by an external alternating magnetic field. The authors successfully demonstrate bi-directional motion of the structure in response to amplitude and frequency of the driving magnetic field, and realized a cargo transport by these new structures.

The work is very detailed and clearly written. It also combines experiments with simulations that make underlying science and conclusions well supported.

Overall, I like the paper and think it will be appreciated by a general reader. I recommend it for publication.

I have a few comments for authors to address though.

1. In Fig 3 where the authors report a frequency dependence of the swimming velocity for different field amplitudes for regimes where nanorod acts as a pusher simulation curves are all "sitting" on top of each other while experimental curves (especially one for 11.3mT) shows much higher velocities than the rest of the curves. What is the reason for that? (I suppose the nanorod – sphere structure used was the same for all curves taken, so it is not due to geometrical differences).
2. On Page 4 authors state that direction of the particle magnetic moment follows the reorientation of the nanorod which induces a rotation of the spherical colloid. Is it indeed enough magnetic anisotropy in those paramagnetic particles that would warrant such mechanical rotation? Do you see the rotation in experiments?
3. I guess in Fig 2(a) instead of B_{c1} it should be B_{c2} since according to the text no self-propulsion happens if $B < B_{c1}$.

Reviewer #3 (Remarks to the Author):

In their work, Garcia-Torres et al. present a self-assembly method leading to magnetic microswimmers. The as such assembled swimmers can assemble and disassemble. The authors might have overlooked certain works such as Cheang et al., PRE 2014, where changes in direction are reported for changes in frequency and Vach et al. Sci. Rep., 2015, where magnetic aggregates assemble to form swimmers. Therefore, this work unifies these two aspects and will deserve publication providing the authors clarify the following points:

- first, it is unclear how effective the process is. Is it easy to form such swimmers? all rods will find their partner particles or only 1% will find each other? what are concentrations needed?

- Do surfaces play a role here? the schemes in figure 2 are unclear. Where is the bottom, what is the "real" geometric configuration? I would even potentially expect a case where magnetic interactions are larger than gravity, and then the rod might "jump" to the other side of the particle.

- Figure 3 a shows the speed / frequency relationship. If the authors assess their propellers change direction at given frequency, there is really only one point where this is seen, the others are not really convincing, and the speed is very limited. Can the authors transform it in a non-dimensional speed? would it be more important? as such, I would rather say the swimmers stop at the given frequency than they swim in the other direction.

- Here, I misused swimmers, which is only used once by the authors, are the "robots" really swimming? or rather rolling?

Finally, the end of the work (chapter cargo transport and miniaturization) rather seems preliminary and need more convincing data to become publishable. There is no proof of functionalization whatsoever, are again the "trains" easy to form?

Answer to the Reviewers:

We sincerely thank the referees for their constructive criticisms, suggestions and for taking the time to review the manuscript. Our point-by-point responses are listed below.

Reviewer #1

The authors in their manuscript "Magnetically tunable bidirectional locomotion of a self-assembled nanorod-sphere propeller" demonstrate and explore in detail a new design for self-assembled and self-propelling colloidal structure that is most probably scalable to the nanoscale. Controller transport at micro and nanoscale is still a formidable challenge and each new strategy brings us a step closer to the robust and adaptive swimming at the micro and nano scales required for emergent technologies such as targeted drug delivery and soft microrobotics.

In their paper authors report realization of a hybrid nanorod-microspher propeller actuated and controlled by an external alternating magnetic field. The authors successfully demonstrate bi-directional motion of the structure in response to amplitude and frequency of the driving magnetic field, and realized a cargo transport by these new structures.

The work is very detailed and clearly written. It also combines experiments with simulations that make underlying science and conclusions well supported.

Overall, I like the paper and think it will be appreciated by a general reader. I recommend it for publication.

Answer: We thank the referee for all her/his positive comments on our manuscript.

I have a few comments for authors to address though.

1. In Fig 3 where the authors report a frequency dependence of the swimming velocity for different field amplitudes for regimes where nanorod acts as a pusher simulation curves are all "sitting" on top of each other while experimental curves (especially one for 11.3mT) shows much higher velocities than the rest of the curves. What is the reason for that? (I suppose the nanorod –sphere structure used was the same for all curves taken, so it is not due to geometrical differences).

Answer: We find from experiments and simulations that the speed of the propeller when the nanorod acts as a pusher is significantly smaller than the speed as a puller. Also, as observed by the referee, the nanorod-sphere propeller used for all measurements in Fig.3 was the same. However, in this particular regime of motion, the experimental data display deviations from the trend observed in the numerical simulation probably due to presence of disorder

in form of surface asperities, or thermal noise that reduce the propulsion efficiency of the pair as compared to the simulations. In any case, we find that the change in direction is robust, as can be appreciated by enlarging the scale of the negative axis in Fig.3. Thus, we add to Fig.3 a small inset showing the negative velocity when the nanorod acts as a pusher. We also comment the deviation of the experimental data from the simulation one in the main text by writing on page 5, column 1:

"We note that, when comparing the two graphs in Fig.3, the experimental data display some deviations from the simulations especially when the nanorod act as a pusher, here characterized by a smaller translational velocity. These deviations can be attributed to the presence of noise in the experimental system, coming e.g. from thermal fluctuations or inhomogeneities in the glass surface and in the magnetic moment of the nanorod. Despite that, in this regime we found a robust trend where the velocity of the ensemble reverses its direction as the frequency increases, as shown in the inset on top of Fig.3."

2. On Page 4 authors state that direction of the particle magnetic moment follows the reorientation of the nanorod which induces a rotation of the spherical colloid. Is it indeed enough magnetic anisotropy in those paramagnetic particles that would warrant such mechanical rotation? Do you see the rotation in experiments?

Answer: The referee asks an interesting question, whether the particle magnetic anisotropy allows to induce the rotation of the paramagnetic colloid. From the experimental videos of the pair colloid-nanorod, it is not possible to directly visualize the particle rotation due to its spherical symmetry. Thus, a first indirect evidence of this rotation comes from the numerical simulation. However, to confirm this point, we have performed further experiments by holding an individual particle to an external rotating magnetic field polarized in the (x,z) plane, i.e. the plane perpendicular to the glass substrate (x,y). We find that the rotating field is able to torque the particle and induce a rotational motion, as confirmed by the presence of a net drift velocity of the particle. We have attached an experimental video (Movie_for_referee.WMV) where the particle is subjected to a rotating field with amplitude $B_0 = 5.2\text{mT}$ and frequency $f = 10\text{Hz}$, thus similar to the one used in the experiments, and displaying a translational speed of $v = 7$ micron/s. This velocity arises from the hydrodynamic rotation/translational coupling between the particle and the plane. We comment this point in the text by writing at the end of page 4 the following:

"The rotational motion of the paramagnetic sphere results from the presence of magnetic anisotropy within the particle. Indeed, in a separate experiment, we have observed that under a rotating field circularly polarized in the (x,z) plane the particle experiences a magnetic torque, and acquires a net translational motion close to the glass surface."

3. I guess in Fig 2(a) instead of B_{c1} it should be B_{c2} since according to the text no self-propulsion happens if $B < B_{c1}$.

Answer: We thank the referee for noticing this typo. Effectively in Fig.2(a) it should be B_{c2} and not B_{c1} . Thus, we have made the corresponding changes in the new version of the manuscript.

Reviewer #3

In their work, Garcia-Torres et al. present a self-assembly method leading to magnetic microswimmers. The as such assembled swimmers can assemble and disassemble. The authors might have overlooked certain works such as Cheang et al., PRE 2014, where changes in direction are reported for changes in frequency and Vach et al. Sci. Rep., 2015, where magnetic aggregates assemble to form swimmers. Therefore, this work unifies these two aspects and will deserve publication providing the authors clarify the following points:

Answer: We thank the referee for her/his positive comment on our manuscript. We also thank for let us know about these works by Cheang and Vach, that are now cited in our manuscript (Ref. 33 and 34).

- first, it is unclear how effective the process is. Is it easy to form such swimmers? all rods will find their partner particles or only 1% will find each other? what are concentrations needed?

Answer: In our experiments we have focused on dilute samples to characterize the dynamics of individual propellers. The magnetic interaction between the nanorods could induce the formation of larger aggregates in more concentrated solutions, given their permanent magnetic moments. Thus, our experiments were performed at relatively low concentrations, and the solutions used contains ~200 nanorods/ml and 200 spheres/ml. Even under our dilute conditions, the nanorod/colloid pair could be easily formed by moving the nanorod towards a paramagnetic colloid using a rotating magnetic field applied perpendicular to the glass substrate, i.e. following the same principle of rotation close to a substrate introduced in [Zhang et al. ACS Nano 2010, 4, 6228 –6234.]. In order to specify the concentration used, we have added in the Method section the following sentence:

"In our system we work under dilute conditions (~200 nanorods/ml and ~200 spheres/ml) in order to focus on the dynamics of isolated propellers."

- Do surfaces play a role here? the schemes in figure 2 are unclear. Where is the bottom, what is the "real" geometric configuration? I would even potentially

expect a case where magnetic interactions are larger than gravity, and then the rod might "jump" to the other side of the particle.

Answer: In our work the presence of the substrate plays a major role in the propulsion of the doublet: the net translation induced by the rotation of the rod takes place due to the asymmetry in frictional force generated by the close proximity with the glass surface. We probably have not specified this point clearly enough in the manuscript. Thus, we write on page 3, column 1 the following:

"As a consequence of its rotational motion, the nanorod translates due to the close proximity of the surface, see the geometry in Fig.2(a). The hydrodynamic interaction with the surface creates an asymmetry in friction that is rectified into a net translational motion [New Ref.]. Thus, the nanorod acts as a puller..."

[New Ref.] Zhang et al. ACS Nano 2010, 4, 6228 –6234

Instead of:

"As a consequence of its rotational motion, the nanorod translates due to the hydrodynamic interaction with the plane, acting as a puller..."

Regarding the interesting situation described by the referee, where the nanorod could "jump" to the other side of the paramagnetic colloid, we did not observe it in the experiments and in the simulation. The reason is that for our geometry, where the diameter of the spherical particle is of the same order as the length of the rod, the nanorod rotational motion is very fast, compared to the time required to jump on the other side of the particle.

- Figure 3 a shows the speed / frequency relationship. If the authors assess their propellers change direction at given frequency, there is really only one point where this is seen, the others are not really convincing, and the speed is very limited. Can the authors transform it in a non-dimensional speed? would it be more important? as such, I would rather say the swimmers stop at the given frequency than they swim in the other direction.

Answer: The speed of the propeller when the nanorod acts as a pusher is significantly smaller than the speed as a puller, but there is a clear measurable translation. We agree with the referee that such speed is not well appreciated in the Fig.3a due to the wide range of the Y-axis scale. To address this issue, we have added in the top graph of Fig.3 a small inset illustrating the corresponding velocity trend as a function of the driving frequency and on an enlarged scale, please see also the answer to Reviewer 1.

We also follow the referee suggestion and normalize the velocity of the nanorod-sphere pair with respect to the field frequency and the length of the nanorod. The corresponding image for the experimental data is attached below and confirm effectively that, by increasing the field the pair changes direction of motion.

- Here, I misused swimmers, which is only used once by the authors, are the "robots" really swimming? or rather rolling?

Answer: The colloid-nanorod pair displays a combination of rotational and translational motion depending on the orientation of the applied field. The nanorod rotates close to the surface in order to follow the field direction, thus it would be similar to a roller if taken individually. However, at the same time, the pair repel or attract due to dipolar interaction, displaying an additional relative translational motion. Thus, we mainly name it propeller in the text to differentiate it from "swimmer" as the long-time dynamics is rather deterministic, with not random switching as observed in biological microswimmers.

Finally, the end of the work (chapter cargo transport and miniaturization) rather seems preliminary and need more convincing data to become publishable. There is no proof of functionalization whatsoever, are again the "trains" easy to form?

Answer: We agree with the referee that the final part of the manuscript describes initial results related with the potential application of our magnetic prototype as cargo transporter. However, the main subject of the work is the demonstration and the corresponding theoretical understanding of a new

transport mechanism based on the fine balance between magnetism, gravity and hydrodynamics. This mechanism shows a rich physical behaviour including the inversion of velocity by varying the amplitude or frequency of the applied field.

Regarding the case of functionalization, we probably did not clearly explain in the text our meaning, i.e. that all the transport features reported in this manuscript are based on sole magnetic interactions, not on chemical functionalization. The potential use of chemical functionalization could add a further degree of functionality to our system. We have rewritten the phrase in the manuscript as:

"Another future possibility would be to chemically functionalize..."

Instead of:

"One can functionalize..."

Finally, regarding the train formation, we have indeed increased the concentration of particles in the experimental solution (~500 spheres/ml) in order to easily form the trains of particles. Also, this process is not difficult as the magnetic nanorods can be easily transported to different locations of the experimental platform close to the paramagnetic particles, where dipolar interactions are used to bind them into elongated structures. We comment this in the Method section by writing:

"However, for the train formation larger quantities of particles were used, ~500 spheres/ml."

REVIEWERS' COMMENTS:

Reviewer #1 (Remarks to the Author):

In the revised version of the manuscript authors successfully addressed all scientific points raised by the Reviewers.

A minor suggestion:

On page 1, first column in a sentence starting with “This situation can be avoided ...” when authors talk about micro-propellers actuated by external fields, including electric field driven (Quincke) rollers, it might be appropriate to also cite recently discovered magnetic field actuated rollers (Science Advances 3, e1601469, 2017).

I am comfortable recommending this manuscript for publication in Nature Communications.

Reviewer #3 (Remarks to the Author):

With the changes made by the authors, the paper should be published now.

Answer to the Reviewers:

We sincerely thank the referees for taking the time to review our manuscript again. Our point-by-point responses are listed below.

Reviewer #1

In the revised version of the manuscript authors successfully addressed all scientific points raised by the Reviewers.

A minor suggestion:

On page 1, first column in a sentence starting with “This situation can be avoided ...” when authors talk about micro-propellers actuated by external fields, including electric field driven (Quincke) rollers, it might be appropriate to also cite recently discovered magnetic field actuated rollers (Science Advances 3, e1601469, 2017).

I am comfortable recommending this manuscript for publication in Nature Communications.

Answer: We thank the referee for all her/his positive comments on our manuscript. We have added the reference in the main text.

Reviewer #3

With the changes made by the authors, the paper should be published now.

Answer: We thank the referee for being in support of our manuscript.